# Effect of Different Landing Heights and Loads on Ankle Inversion Proprioception during Landing in Individuals with and without Chronic Ankle Instability

**DOI:** 10.3390/bioengineering9120743

**Published:** 2022-11-30

**Authors:** Ming Kang, Tongzhou Zhang, Ruoni Yu, Charlotte Ganderton, Roger Adams, Jia Han

**Affiliations:** 1School of Exercise and Health, Shanghai University of Sport, Shanghai 200438, China; 2School of Medicine, Jinhua Polytechnic, Jinhua 321000, China; 3Faculty of Health, Arts and Design, Swinburne University of Technology, Hawthorn, VIC 3122, Australia; 4Research Institute for Sport and Exercise, University of Canberra, Canberra, ACT 2234, Australia; 5College of Rehabilitation Sciences, Shanghai University of Medicine and Health Sciences, Shanghai 201318, China

**Keywords:** proprioception, ankle sprain, chronic ankle instability, landing

## Abstract

Proprioception is essential for neuromuscular control in relation to sport injury and performance. The effect of landing heights and loads on ankle inversion proprioceptive performance in individuals with or without chronic ankle instability (CAI) may be important but are still unclear. Forty-three participants (21 CAI and 22 non-CAI) volunteered for this study. The Ankle Inversion Discrimination Apparatus for Landing (AIDAL), with one foot landing on a horizontal surface and the test foot landing on an angled surface (10°, 12°, 14°, 16°), was utilized to assess ankle proprioception during landing. All participants performed the task from a landing height of 10 cm and 20 cm with 100% and 110% body weight loading. The four testing conditions were randomized. A repeated measures ANOVA was used for data analysis. The result showed that individuals with CAI performed significantly worse across the four testing conditions (*p* = 0.018). In addition, an increased landing height (*p* = 0.010), not loading (*p* > 0.05), significantly impaired ankle inversion discrimination sensitivity. In conclusion, compared to non-CAI, individuals with CAI showed significantly worse ankle inversion proprioceptive performance during landing. An increased landing height, not loading, resulted in decreased ankle proprioceptive sensitivity. These findings suggest that landing from a higher platform may increase the uncertainty of judging ankle positions in space, which may increase the risk of ankle injury.

## 1. Introduction

Ankle sprain is a common sports injury [1] that usually occurs during jump and landing activities [2,3,4,5] in a foot-inversion position [6,7] and may lead to chronic ankle instability (CAI) [8]. Proprioception is fundamental for neuromuscular control in ankle injury, and can be defined as the ability of an individual to integrate sensory signals from mechanoreceptors to perceive the location and spatial movement of body parts [9]. Studies have shown that neuromuscular control is deficient and ankle inversion proprioception to be significantly impaired and that the lower limb proximal muscle activity pattern is altered in individuals with CAI [10,11], especially during a landing task [12]. In some sports, such as cross-country running [13], participants may carry weights and land from different heights. It is unknown to what extent these factors may affect ankle proprioception during jump landing. Thus, exploring ankle inversion proprioception during landing is essential to understand the sensorimotor mechanisms underlying ankle sprains, and may inform prevention and rehabilitation of ankle sprain.

Previous biomechanical studies [14,15] suggests that significant biomechanical changes in movement patterns, muscle activation, and muscle mechanics occur in lower extremities when landing at different landing heights. Wang [16] reported that as landing height increases, angular displacement of the ankle, knee, and hip joints also increases, and lower extremity injuries are more likely to occur during landing. In addition, higher landing heights may lead to an increase in the velocity of the foot on landing, which is highly susceptible to injury of the foot and ankle complex [17], especially given that individuals with CAI have a delayed response of valgus muscle [18]. Fong [19] reported that within 0.11 s after the foot strike the ankle is in a position where it could be injured. Although these empirical studies indicate that motor behavior alters at different landing heights, it remains unclear whether the perceptual systems will also change. Although most current proprioceptive testing methods lack the ecological validity to carry out proprioceptive assessment during landing [9], Han developed the Ankle Inversion Discrimination Apparatus for Landing (AIDAL) [12] to make this measurement task achievable. We speculate whether the height of the landing affects the proprioceptive system, which further affects ankle stability.

Load is known to affect perceptual and motor system performance by altering postural control and proprioception. For instance, extra body weight can reduce postural-stability control and the sensory pathways from the foot sole [20,21]. However, weight-bearing can improve proprioceptive performance. The proprioceptive sensitivity of the knee joint was significantly better in full weight-bearing conditions than partial weight-bearing of the lower limb [22]. Considering the crucial significance of proprioception for postural control [23,24], we wondered whether the excess body load would have the effect on ankle proprioception during landing.

CAI is a condition characterized by pain, weakness, reduced ankle range of motion, perceived ankle “giving way” sensation, and proprioception deficit, which may lead to recurrent ankle sprains [25,26,27,28]. Although the soft tissues of the foot (e.g., fascia, muscles) perform a crucial role in maintaining the ankle stability, recurrent ankle sprains disrupt the original structure. Studies have shown that the thickness of the plantar fascia is reduced after lateral ankle sprain [29]. Moreover, individuals with CAI demonstrate sensorimotor insufficiencies and proprioception deficits [10,30], as well as biomechanical variations in lower limb movement patterns, motor strategies, muscle activation, and leg stiffness control during landing [24,31,32,33,34]. However, it is unclear whether a change in landing height and loading may have different effects on individuals with and without CAI. Evidence has suggested that the mechanoreceptors around the ankle joint are likely to be damaged in CAI [35], and so patients had diminished in neuromuscular control, especially ankle proprioception [5,10,12]. In addition, recent neuroimaging studies have found that this specific population also shows central change when performing proprioceptive balance control task [36,37,38]. These findings suggest that individuals with CAI may perform differently compared to their non-CAI counterparts.

Accordingly, the aim of this study was to test the ankle inversion proprioception of individuals with and without CAI when they landed at different heights with different loads. We hypothesized that an increased landing height may reduce ankle proprioception, but extra loads may reverse this, and individuals with CAI would perform significantly worse than those without CAI.

## 2. Materials and Methods

### 2.1. Study Design

A cross-sectional study was performed between June and July 2021 according to the Strengthening the Reporting of Observational Studies in Epidemiology (STROBE) recommendations.

### 2.2. Participants

Forty-three participants were recruited (21 CAI and 22 non-CAI). To be eligible, the CAI participants must have had the following: (i) at least one ankle sprain that caused an inflammatory reaction (pain, swelling, etc.) in the previous 12 months; (ii) at least two episodes of ankle instability “giving way” or repeated sprains within 6 months before the test; (iii) a Cumberland Ankle Instability Tool score (CAIT) [39] of <24; and (iv) not have had an ankle injury within 3-months of being tested. These inclusion criteria align with the recommendations of the International Ankle Consortium [39]. The non-CAI participants had no subjective reports of ankle instability, recurrent ankle sprains, and neurological or motor dysfunction. All participants were excluded from the study if they had a history of lower limb surgery, fracture, or any acute injury of other joints of the lower limb in the 3-months prior to the commencement of the study. The latest ankle sprain was 15 weeks on average before the test for the CAI group.

The study was conducted in accordance with the Declaration of Helsinki and was approved by the Shanghai University of Sport Ethics Committee (102772021RT073). Written informed consent was obtained from participants before data collection.

### 2.3. Apparatus

This study utilized the AIDAL to measure the acuity of ankle inversion proprioception during landing [12]. This method has shown good test–retest reliability and validity in distinguishing individuals with and without CAI [40]. The AIDAL (Figure 1) consists of three parts: the take-off platform (A/D), the horizontal landing platform for the supporting foot (B), and the tilted landing platform for the testing foot (C). The four different angles of ankle inversion were generated by 4 wedged landing platforms: inversion 1 = 10°, inversion 2 = 12°, inversion 3 = 14°, and inversion 4 = 16°. The landing heights were 10 cm (A to B) and 20 cm (D to B) (Figure 1 and Figure 2, a and b).

Before the AIDAL’s data collection, each participant had three rounds of familiarization of the 4 possible ankle inversions in order (12 trials in total), and they were required to remember the four different ankle inversions during the familiarization session. Participants then undertook 40 trials of testing, with 10 for each inversion position presented randomly. Participants were required to make an absolute judgment about the ankle inversion angle on each testing trial, and the numbers (either 1, 2, 3, or 4) with which the participant responded were collected, and without feedback being given as to the correctness of their judgements.

The weight-adjustable vest (Figure 2c,d) was used to provide 10% of extra body weight during the AIDAL proprioception test [22].

### 2.4. Procedures

The experiment was conducted in a university laboratory. All participants were tested with bare feet. For the non-CAI group, the testing foot was randomly selected, and for the CAI group, with unilateral CAI (n = 10), we tested the affected ankle, and with bilateral CAI (n = 11), we tested the ankle with a lower CAIT score. The flow chart of this study is shown below (Figure 3). All participants were tested from a jumping height of 10 cm and 20 cm without extra load (100% body weight), and with an extra 10% of body weight (110% body weight) (Figure 2). The four testing conditions were randomized with a 15-min break between testing sessions. During the test, participants were instructed to keep their head and eyes forward to eliminate visual information about the landing platforms. A single examiner who was blinded to participants’ ankle stability status performed all experiments.

### 2.5. Data Analysis

SPSS version 25 (Armonk, NY, USA) was used for data analysis and a *p* value of 0.05 or less was used to determine statistical significance. A total of 40 presentations of ankle inversion and related participant responses were inputted into SPSS to generate a receiver operating curve (ROC), and the area under the curve (AUC) was calculated as the ankle proprioception discrimination sensitivity. The AUC value ranges between 0.5 and 1. A higher value represents more accurate proprioception sensitivity.

Given that the data for the two groups were normally distributed, to determine the effect of CAI, landing heights and loads on ankle proprioception, a repeated measures ANOVA was conducted on the AUC scores.

## 3. Results

Demographic information of the included participants is reported in Table 1. The repeated measures ANOVA showed CAI and the landing height main effects. Specifically, the CAI group performed significantly worse than the non-CAI group (F = 6.120, *p* = 0.018, partial *η*^2^ = 0.130) and the ankle proprioception AUC scores of 20 cm landing height were significantly lower than that of 10 cm landing height (F = 7.216, *p* = 0.010, partial *η*^2^ = 0.150) (Table 2 and Figure 4). However, there was not a significant load main effect (F < 0.001, *p* = 0.995, partial *η*^2^ < 0.001) or an interaction effect of height, load, and presence of CAI (Table 2).

## 4. Discussion

Consistent with our hypothesis, we found that the overall ankle inversion proprioceptive performance of CAI patients was significantly worse than that of non-CAI participants, suggesting that CAI patients have impaired proprioceptive control when landing on an uneven surface. In addition, ankle proprioceptive discrimination sensitivity was significantly worsened by an increased landing height, but not loading (Figure 4), which was true for both CAI and non-CAI groups.

Our results showed that the proprioceptive acuity of CAI participants was significantly worse than those without CAI across the four different testing conditions. These findings are consistent with prior studies [12,41] and further supports the notion that CAI patients have impaired somatosensory control during landing on an inverted ankle [12]. Some studies have shown that individuals with CAI exhibit altered peak proximal muscle forces, force-generating capacities, as well as greater hip flexion and ankle inversion angles, and peak vertical ground reaction forces during landing tasks [42,43,44]. Our findings complement the substantial deviations in the lower limb motor output observed between CAI and non-CAI individuals and have shown that ankle proprioceptive input is also different between the two groups. According to the research of Waddington and Adams [45], even a 0.04° increase in inversion uncertainty has the potential to raise the probability of injury when landing on the inverted ankle from 1.2% to 1.22%. Although this 0.02% increase in injury seems low, it could become a significant influence in the occurrence of injury due to the fact that landings are numerous in sports activities. Therefore, the difference in proprioception between the CAI and non-CAI participants in this study has significant implications for ankle stability and may raise the risk of sprain. Given that ankle proprioception is fundamental for lower limb motor control [23], the difference in proprioceptive performance during landing found here may partially explain the motor output difference between CAI and non-CAI observed in previous studies [43,44]. Future research may explore if any rehabilitation program that targets ankle proprioception [46] could have positive effects on lower limb motor control in people with a post-ankle sprain that may or may not develop CAI.

In terms of the effect of landing height on lower limb biomechanics, research has shown that with increasing landing height, the ankle dorsiflexion, knee extension, and peak ankle plantarflexion moments are significantly altered in individuals with CAI [47]. The results of the current study show that an increased landing height could impair ankle inversion proprioception in both CAI and non-CAI participants. This evidence supports the notion that the larger the movement amplitude, the worse the proprioceptive performance [48], suggesting that landing from a higher place may be associated with increased noise in the perceptual systems of brain [48,49], so that ankle inversion movement discrimination sensitivity is decreased. Previous upper [48,50] and lower [51] limb proprioception studies have found that larger movement resulted in worse proprioceptive acuity. This is consistent with the findings of our result. A greater movement amplitude would generate more noise and increased uncertainty when judging limb positions in space. This finding suggests that higher jump–landing may increase the risk of ankle injury and partially explains why landing from a jump is one of the most common mechanisms for ankle injury [2], especially in ankle joint inversion landing conditions [52].

However, we found an extra 10% of body weight for the proprioceptive measurement did not differ from 100% body weight conditions. This evidence is contrary to the argument that ankle proprioception simply relied on mechanoreceptors around the foot–ankle complex [53]. Han et al. [9,54] argue that proprioception assessment methods can be classified into testing passively “imposed” and actively “obtained” proprioception. The imposed methods [53] believed that proprioception is completely reliant on information received from the peripheral proprioceptors mechanoreceptors (i.e., muscles, joints, and skin) [55]. In contrast, the actively obtained methods [56] were developed on the basis of an ecological validity concept that proprioceptive performance is not fully determined by passively-imposed proprioceptive signals from mechanoreceptors, but requires adaptive central processing of multiple sources of information [9]. If the “imposed” proprioception view is true, then both an increased landing height and load could further activate mechanoreceptors located around the foot–ankle complex, improving ankle proprioception. However, the results here did not support this notion. The possible reason is that an additional 10% of load may not have been large enough and that the central nervous system may have mechanisms to adapt to the noise generated by mild changes in weight-bearing conditions. In addition, although we required participants to land evenly, participants might not have achieved a balanced distribution of weight between the two feet, with more weight on the horizontal platform, thus this may have reduced the load on the inverted foot.

This study compared ankle inversion proprioception during landing in both CAI and non-CAI populations in the face of height and loading conditions. One of the limitations of this study was that we did not collect data about the physical activity and occupation of the participants included in this study, which may have an impact on the results. Furthermore, we did not quantify kinematic patterns so that precise changes in movement extent of the participants at different heights and loading states to be observed. We did not use any instruments to control whether participants had a balanced distribution of weight between the two feet, a feature which should be improved for future study. In addition, previous research has shown that female and male individuals with CAI performed differently on a range of functional tasks [57]. However, given the relatively small sample size of the current study, the sex differences in this proprioceptive task were not revealed. Furthermore, the participants involved in the current study were relatively young and it is unknown if the findings here can be generalized to other age groups.

## 5. Conclusions

Compared to non-CAI, individuals with CAI showed significantly worse ankle inversion proprioceptive performance during landing. An increased landing height, but not loading, resulted in decreased ankle proprioceptive sensitivity. These findings suggest that landing from a higher place may increase the uncertainty of judging ankle positions in space, and thus increases the risk of ankle injuries. Therefore, jump-landing exercise from different heights may be important for ankle injury prevention and rehabilitation. Future research may investigate the effects of a rehabilitation program targeting ankle inversion proprioception during landing and explore the peripheral and central mechanisms associated with these effects.

## Figures and Tables

**Figure 1 bioengineering-09-00743-f001:**
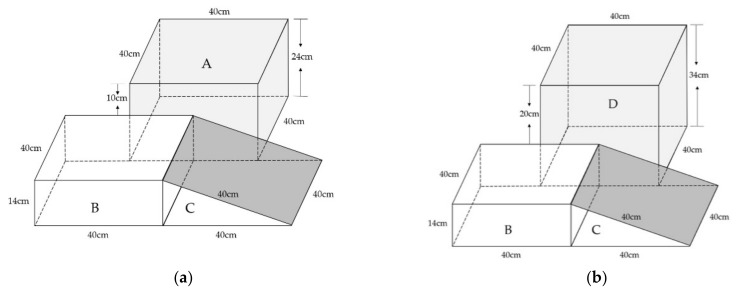
The two different heights of AIDAL: (**a**) 10 cm heights between A and B; (**b**) 20 cm heights between D and B.

**Figure 2 bioengineering-09-00743-f002:**
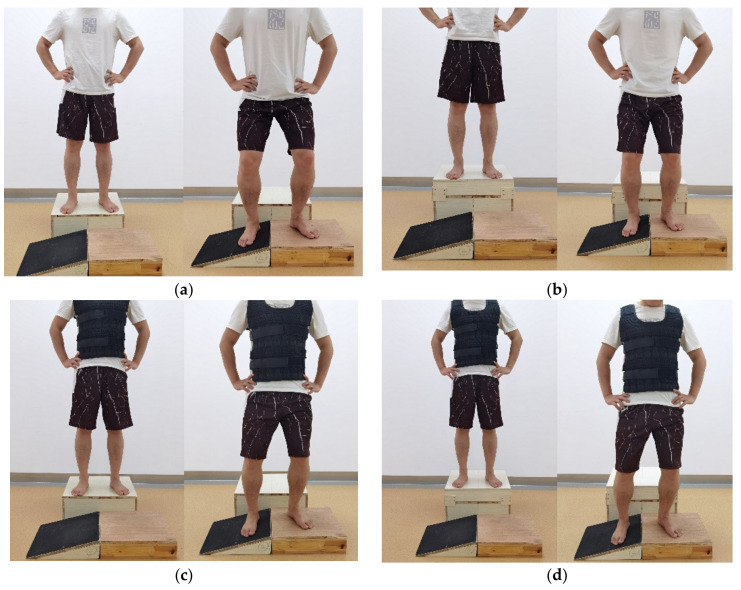
The different landing conditions. (**a**) load 100% and 10 cm heights; (**b**) load 100% and 20 cm heights; (**c**) load 110% and 10 cm heights; (**d**) load 110% and 20 cm heights.

**Figure 3 bioengineering-09-00743-f003:**
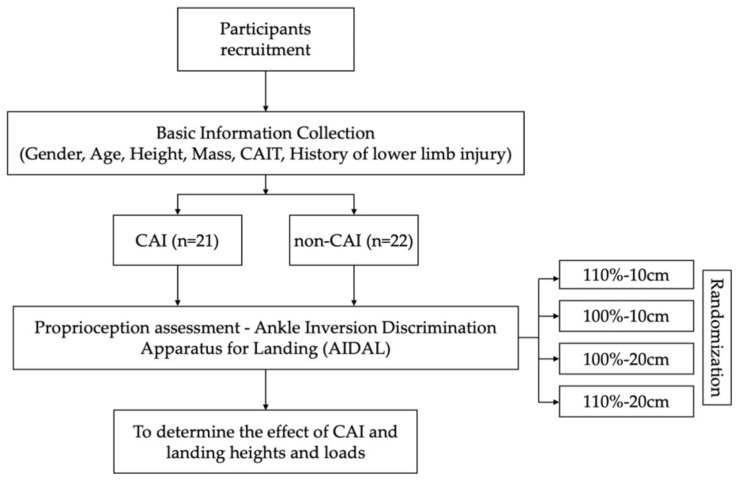
The flow chart of this study. A total of 100% meant original body weight and 110% meant 10% extra body weight; 10 cm and 20 cm indicated landing heights.

**Figure 4 bioengineering-09-00743-f004:**
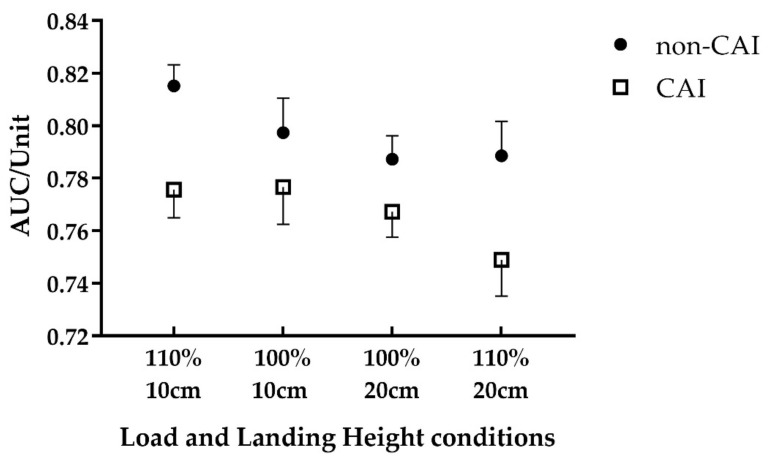
Differences in ankle inversion discrimination tested during landing heights and loads between individuals with and without CAI. A total of 100% indicated original body weight and 110% indicated 10% extra body weight; 10 cm and 20 cm represented landing heights. Compared to non-CAI, CAI showed significantly worse ankle inversion proprioceptive performance during landing (F = 6.120, *p* = 0.018), and the AUC score decreased by increased landing height for all participants (F = 7.216, *p* = 0.010).

**Table 1 bioengineering-09-00743-t001:** Participant demographic information (Mean ± SD).

Characteristic	Group		Difference between Groups
	CAI	Non-CAI	
N	21	22	-
Gender	M10 F11	M11 F11	-
Age (y)	23.4 ± 3.2	24.1 ± 2.1	*t* = −0.804, *p* = 0.426
Height (cm)	171.3 ± 8.2	169.1 ± 6.3	*t* = 0.972, *p* = 0.337
Mass (kg)	65.6 ± 11.6	64.6 ± 9.0	*t* = 0.268, *p* = 0.790
CAIT score	15.6 ± 4.9	28.6 ± 1.8	*t* = −11.606, *p* = 0.000

SD = standard deviation, CAI = chronic ankle instability, N = Number, M/F = male/female, CAIT = Cumberland Ankle Instability Tool.

**Table 2 bioengineering-09-00743-t002:** A repeated measures ANOVA of the average AUC scores for landing height and group.

	F	*p*	Partial *η*^2^
10 cm vs. 20 cm	7.216	0.010 *	0.150
CAI vs. Non-CAI	6.120	0.018 *	0.130
100% vs. 110%	<0.001	0.995	<0.001
Height × Load	1.874	0.178	0.044
Height × CAI	<0.001	0.984	<0.001
Load × CAI	2.244	0.142	0.052
Height × Load × CAI	0.001	0.972	<0.001

× = interaction between loads and heights. * = *p* < 0.05 between groups.

## Data Availability

The data presented in this study are available on request from the corresponding author. The data are not publicly available due to ethical or privacy restrictions.

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
