# Peer review of "Effect of Different Landing Heights and Loads on Ankle Inversion Proprioception during Landing in Individuals with and without Chronic Ankle Instability"

_bioengineering, 2022, doi:10.3390/bioengineering9120743_

Round 1
Reviewer 1 Report
The rationale of the study is presented vaguelly. It should be improved and revised.
More details about subjects characteristics are needed - the physical activity level, time from the last ankle sprain. Were the footballers n this group ?
The Ankle inversion discrimination apparatus for landing (AIDAL) was used by Authors, but this method was not sufficiently described in the methods. What data were collected and analysed? How the propiroception was assessed? Much more details is needed.
The testng procrdure should be describe better - was the landing tasks performed randomly? Were the subjects familiarised? Did they perform any testing trials before measurements? Hpw many trials were done?
The discussion is to much descriptive, to long and in many parts beyonde the scope of this study without any reference to the results of own research presented in this work. The own results should be discussed in conjunction with the results of other authors data. I recommend that authors should be more precise and focused on their own results. The discussion should be revised,, and much more emphesized on Authors own results
Author Response
Response letter
Dear Editor,
Thank you for sending reviewer comments, and thank you for the opportunity to revise this manuscript. We have addressed each reviewer point and revised the manuscript accordingly. We hope that the manuscript is now satisfactory and ready for publication in the Bioengineering.
Response to Reviewer 1 Comments
Point 1: The rationale of the study is presented vaguely. It should be improved and revised.
Response 1: Thank you. We have now revised the manuscript throughout to make the study rationale clearer.
Point 2: More details about subjects characteristics are needed - the physical activity level, time from the last ankle sprain. Were the footballers n this group?
Response 2: Thank you for your comment. To keep our questions of the athletes to a minimum, we did not collect data about physical activity level. Therefore, we have added this point in the Limitation section. (See page 7, line 336-339)
Line 336-339:
“One of the limitations of this study was that we did not collect the data about the physical activity of the participants included in this study, which may have an impact on the results. Furthermore,”
We have added the information about the time from the last ankle sprain in the “2.2 Participants” (See page 3, line 138-139)
Line 138-139:
“The latest ankle sprain was 15 weeks in average before the test for the CAI group.”
Point 3: The Ankle inversion discrimination apparatus for landing (AIDAL) was used by Authors, but this method was not sufficiently described in the methods. What data were collected and analysed? How the propiroception was assessed? Much more details is needed.
Response 3: Thank you. we have now added more detailed information about the AIDAL and assessment.
We added “and the numbers (either 1, 2, 3, or 4) with which the participant responded were collected,” in front of the “without feedback being given as to the correctness of their judgements.” (See page 4, line 172-175)
“Participants were required to make an absolute judgment about the ankle inversion angle on each testing trial, and the numbers (either 1, 2, 3, or 4) with which the participant responded were collected, without feedback being given as to the correctness of their judgements.” (Line 172-175)
“40 presentations of ankle inversion and related participant responses were input into SPSS to generate a Receiver Operating Curve (ROC), and the area under the curve (AUC) was calculated as the ankle proprioception discrimination sensitivity. The AUC value ranges between 0.5 and 1. A higher value represents more accurate proprioception sensitivity.” (Line 197-201)
Point 4: The testing procedure should be describe better - was the landing tasks performed randomly? Were the subjects familiarised? Did they perform any testing trials before measurements? How many trials were done?
Response 4: Thank you., We have clarified the details about the testing procedure. (See page 4, line 168-175, 184-185)
Line 168-175:
“Before the AIDAL data collection, each participant had three rounds of familiarization of the 4 possible ankle inversions in order (12 trials in total), and they were required to remember the four different ankle inversions during the familiarization session. Participants then undertook 40 trials of testing, with 10 for each inversion position presented randomly. Participants were required to make an absolute judgment about the ankle inversion angle on each testing trial, and the numbers (either 1, 2, 3, or 4) with which the participant responded were collected”
Line 184-185:
“The four testing conditions were randomized with a 15-minute break between testing sessions.”
Point 5: The discussion is to much descriptive, to long and in many parts beyonde the scope of this study without any reference to the results of own research presented in this work. The own results should be discussed in conjunction with the results of other authors data. I recommend that authors should be more precise and focused on their own results. The discussion should be revised, and much more emphasized on Authors own results
Response 5: Thank you for your suggestion. We have now revised the discussion and emphasized on our own results.
“4. Discussion
Consistent with our hypothesis, we found that the overall ankle inversion proprioceptive performance of CAI patients was significantly worse than that of non-CAI participants, suggesting that CAI patients have impaired proprioceptive control when landing on uneven surface. In addition, ankle proprioceptive discrimination sensitivity was significantly worsened by increased landing height, but not loading (Figure 4), which was true for both CAI and non-CAI groups.
Our results showed that the proprioceptive acuity of CAI participants was significantly worse than those without CAI across the 4 different testing conditions. These findings are consistent with prior studies [12,41] and further supports the notion that CAI patients have impaired somatosensory control during landing on inverted ankle [12]. Some studies have shown that individuals with CAI exhibit altered peak proximal muscle forces, force-generating capacities, as well as greater hip flexion and ankle inversion angles and peak vertical ground reaction forces during landing tasks [42-44]. Our findings complement the substantial deviations in the lower limb motor output observed between CAI and non-CAI individuals and have shown that ankle proprioceptive input is also different between the two groups. According to Waddington and Adams' research [45], even a 0.04° increase in inversion uncertainty has the potential to raise the probability of injury when landing on the inverted ankle from 1.2% to 1.22%. Although this 0.02% increase in injury seems low, it could become a significant influence in the occurrence of injury due to the fact that landings are numerous in sports activities. Therefore, the difference in proprioception between the CAI and non-CAI participants in this study has significant implications for ankle stability and may raise the risk of sprain. Given that ankle proprioception is fundamental for lower limb motor control [23], the difference in proprioceptive performance during landing found here may partially explain the motor output difference between CAI and non-CAI observed in previous studies [43,44]. Future research may explore if any rehabilitation program targets ankle proprioception [46] may have positive effects on lower limb motor control in people post ankle sprain that may or may not develop CAI.
In terms of the effect of landing height on lower limb biomechanics, research has shown that with increasing landing height, the ankle dorsiflexion, knee extension and peak ankle plantarflexion moments are significantly altered in individuals with CAI [47]. The results of the current study show that increased landing height could impair ankle inversion proprioception in both CAI and non-CAI participants. This evidence supports the notion that the larger movement amplitude, the worse proprioceptive performance [48], suggesting that landing from a higher place may be associated with increased noise in the perceptual systems of brain [48,49] so that ankle inversion movement discrimination sensitivity is decreased. Previous upper [48,50] and lower [51] limb proprioception studies have found that larger movement resulted in worse proprioceptive acuity. These are consistent with the findings of our result. A greater movement amplitude would generate more noise and increased uncertainty when judging limb positions in space. This finding suggests that higher jump-landing may increase the risk of ankle injury and partially explains why landing from a jump is one of the most common mechanisms for ankle injury [2], especially in the ankle joint inversion landing conditions [52].
However, we found extra 10% of body weight for the proprioceptive measurement did not differ from 100% body weight conditions. This evidence is contrary to the argument that ankle proprioception simply relied on mechanoreceptors around the foot-ankle complex [53]. Han et al. [9,54] argue that proprioception assessment methods can be classified into testing passively “imposed” and actively “obtained” proprioception. Imposed methods [53] believed that proprioception is completely reliant on information received from the peripheral proprioceptors mechanoreceptors (i.e. muscles, joints, and skin) [55]. In contrast, actively obtained methods [56] were developed on the bases of ecological validity concept that proprioceptive performance is not fully determined by passively imposed proprioceptive signals from mechanoreceptors, but requires adaptive central processing of multiple sources of information [9]. If the “imposed” proprioception view is true, then both increased landing height and load could further activate mechanoreceptors located around the foot-ankle complex, improving ankle proprioception. However, the results here did not support this notion. The possible reason is that additional 10% of load may not have been large enough, and the central nervous system may have mechanisms to adapt to the noise generated by mild changes in weight-bearing condition. In addition, although we required participants to land evenly, participants might not have achieved a balanced distribution of weight between the two feet, with more weight on the horizontal platform, thus this may have reduced the load on the inverted foot.
This study compared ankle inversion proprioception during landing in both CAI and non-CAI populations in the face of height and loading conditions. One of the limitations of this study was that we did not collect data about the physical activity and occupation of the participants included in this study, which may have an impact on the results. Furthermore, we did not quantify kinematic patterns so that precise changes in movement extent of the participants at different heights and loading states to be observed. We did not use any instruments to control whether participants had a balanced distribution of weight between the two feet, a feature which should be improved for future study. In addition, previous research has shown that female and male individuals with CAI performed differently on a range of functional tasks [57]. However, given relatively small sample size of the current study, the sex differences in this proprioceptive task were not revealed. Furthermore, the participants involved in the current study were relatively young and it is unknown the findings here can be generalized to other age groups.”

Reviewer 2 Report
General Comments:
The goal of this investigation is similar to the title of the manuscript. The investigators used the AIDAL protocol to investigate inversion when landing. This protocol has subjects land with one foot on a horizontal surface and one foot on an inclined surface. The same foot always landed on the flat or incline. As such, this brings up some limitations that were not addressed by the authors. Since subjects always knew which foot was landing on the incline they could protect against this. As such, this might skew the results, and may be why they didn’t find differences when the extra weight was added. This needs to be discussed and considered.
There are quite a few small errors in English language. I did not point them all out. A careful read by someone more fluent in English should be performed to help them correct these. Otherwise, I have a few additional specific comments that should be addressed below.
I did not examine the supplementary files, they were not referenced from the manuscript.
Specific Comments:
Abstract
11) Line 17, Period needed at end of first sentence.
22) Line 17-18, being unknown is not a reason for performing a study. Would be better to provide rationale in the first two sentences that includes why it is important to understand your topic.
33) Line 19-20, should capitalize all first letters in words forming the AIDAL acronym.
44) For those not fully familiar with the AIDAL, there should be indication in the abstract that one foot landed on a horizontal surface and the test foot on an angled surface ranging from 10-16 degrees.
55) Line 23-24, confusing, you say CAI performed worse in all 4 conditions, but not loading. But loading is 2 of the 4 conditions in the way that you have written the sentence. This sentence should be clarified.
Introduction
11) Line 39-40, what type of cross-country race are you referring to? Maybe a specific example. Any evidence that ankle sprains are high in this type of race?
Materials and Methods
11) Lines 88-94 are manuscript instructions that were not removed by the authors.
22) You describe the requirements of CAI, what were the requirements of the non-CAI subjects?
33) Any constraints on age, BMI, occupation, sex, etc.? It seems you have about half men/women and it was a relatively young adult group.
44) Lines 125-126, confusing, did they perform both unilateral and bilateral landings?
55) Did subjects always know which foot was going to land on the angled platform? If I understand the protocol correctly, the angle was randomized, altered from trial to trial, but never from side to side. Seems like this could introduce some anticipation bias if they always knew which side had the angled surface.
66) How were the subjects blinded to the angle, since it appears they were allowed to have their eyes open. I don’t think asking them to look forward would be enough. I’m also guessing that they knew which height they were dropping from since it looks like this was altered by adding a spacer to the platform they stood on.
77) Were subjects instructed to land with equal weight on each foot? How was this controlled?
88) Was the examiner blinded to height or load? Might they be biased by knowing the conditions?
99) Line 149, were data determined to be normally distributed prior to performing an ANOVA?
Results
11) Table 1, kg is a unit of mass, not weight.
22) Are Table 2 and Figure 3 presenting the same data? It should only be presented in one form. The figure is easier to visualize the observed trends and differences.
Discussion
13) Line 190, confusing, what difference observed in previous studies are you referring to?
24) Line 214-231, in your discussion of the lack of a difference between the two loads, since they knew which foot was landing on the angle, it is possible that they adjusted by landing with more of their weight on the horizontal platform. If this was the case, it could explain the lack of differences.
35) Line 239, did you even run statistics looking for differences between sexes?
46) Seems to me that there are a lot more limitations than you list. For example, if they weren’t blinded to height or foot landing on the angle, this would potentially skew your results. It does not appear that you had a means to measure bodyweight distribution between the feet upon landing to determine if they favored the limb that landed on the angle. The person evaluating the landings did not appear to be blinded to height or load condition.
57) Line 240, Was the fact that you had a relatively young population by design? Similarly, did you recruit to ensure a similar number of men and women? How were the subjects recruited?
Conclusions
18) You found significant effects of group and drop height. Can you comment on whether the magnitude and/or values obtained are clinically significant and if the differences alter their true risk for an ankle injury?
Author Response
Response letter
Dear Editor,
Thank you for sending reviewer comments, and thank you for the opportunity to revise this manuscript. We have addressed each reviewer point and revised the manuscript accordingly. We hope that the manuscript is now satisfactory and ready for publication in the Bioengineering.
Response to Reviewer 2 Comments
Abstract:
Point 1: Line 17, Period needed at end of first sentence.
Response 1: Thank you. We have added a period at the end of the first sentence.
Point 2: Line 17-18, being unknown is not a reason for performing a study. Would be better to provide rationale in the first two sentences that includes why it is important to understand your topic.
Response 2: Thank you for your suggestion. We have changed “is unknown” to “may be important, but are still unclear.” (See page 1, line18-19)
Line18-19:
“The effect of landing heights and loads on ankle inversion proprioceptive performance in individuals with or without chronic ankle instability (CAI) may be important, but are still unclear.”
Point 3: Line 19-20, should capitalize all first letters in words forming the AIDAL acronym.
Response 3: Thank you. We have now changed the format to capitalize the first letters. (See page 1, line20)
Line20:
“The Ankle Inversion Discrimination Apparatus for Landing (AIDAL) was utilized to assess ankle proprioception during landing.”
Point 4: For those not fully familiar with the AIDAL, there should be indication in the abstract that one foot landed on a horizontal surface and the test foot on an angled surface ranging from 10-16 degrees.
Response 4: Thank you. We have added “with one foot landing on a horizontal surface and the test foot landing on an angled surface (10°, 12°, 14°, 16°)” after the “The Ankle Inversion Discrimination Apparatus for Landing (AIDAL)” (See page 1, line20-21).
Line20-21:
“The Ankle Inversion Discrimination Apparatus for Landing (AIDAL) with one foot landing on a horizontal surface and the test foot landing on an angled surface (10°, 12°, 14°, 16°) was utilized to assess ankle proprioception during landing.”
Point 5: Line 23-24, confusing, you say CAI performed worse in all 4 conditions, but not loading. But loading is 2 of the 4 conditions in the way that you have written the sentence. This sentence should be clarified.
Response 5: Thank you. To make it clear, we added the “In addition” in front of the “increased landing height”. (See page 1, line25-27).
Line25-27:
“The result showed individuals with CAI performed significantly worse across the 4 testing conditions (p = 0.018). In addition, increased landing height (p = 0.010), not loading (p > 0.05), significantly impaired ankle inversion discrimination sensitivity.”
Introduction:
Point 1: Line 39-40, what type of cross-country race are you referring to? Maybe a specific example. Any evidence that ankle sprains are high in this type of race?
Response 1: Thanks for your suggestion. We have changed the “race” to “running”, and added the reference in the revised manuscript. (Reference: Hunt, K.J.; Hurwit, D.; Robell, K.; Gatewood, C.; Botser, I.B.; Matheson, G. Incidence and Epidemiology of Foot and Ankle Injuries in Elite Collegiate Athletes. The American journal of sports medicine 2017, 45, 426-433, doi:10.1177/0363546516666815.). (See page 1, line43).
Line43:
In some sports, such as cross-country running [13], participants may carry weights and land from different heights.
Materials and Methods:
Point 1: Lines 88-94 are manuscript instructions that were not removed by the authors.
Response 1: Thank you. We have now removed the instructions.
Point 2: You describe the requirements of CAI, what were the requirements of the non-CAI subjects?
Response 2: Thank you. We have added the “The non-CAI participants had no subjective reports of ankle instability, recurrent ankle sprains, neurological or motor dysfunction.” after the “These inclusion criteria align with the recommendations of the International Ankle Consortium [38].”, and added the “All” before the “participants were excluded from the study if they had history of lower limb surgery, fracture”. It now reads: (See page 3, line134-136).
Line134-136:
“The non-CAI participants had no subjective reports of ankle instability, recurrent ankle sprains, neurological or motor dysfunction. All participants were excluded from the study if they had history of lower limb surgery, fracture;”
Point 3: Any constraints on age, BMI, occupation, sex, etc.? It seems you have about half men/women and it was a relatively young adult group.
Response 3: We limited the age of our participants to 18-30 years when recruiting subjects to ensure that their proprioception would not be affected by the factor of age. As this study was conducted primarily within the university and we tried to keep the number of men and women equal when recruiting subjects, the group tended towards young adults and half men/women.
We have added this point in the Limitation section. (See page 7, line 336-339)
“One of the limitations of this study was that we did not collect data about the physical activity and occupation of the participants included in this study, which may have an impact on the results. Furthermore,”
Point 4: Lines 125-126, confusing, did they perform both unilateral and bilateral landings?
Response 4: The participants were tested on the same ankle throughout the experiment. To make it clear, we have changed “unilateral CAI (n=10) tested the affected ankle, and the ankle with a lower CAIT score was tested in bilateral CAI (n=11).” to “with unilateral CAI (n=10) we tested the affected ankle, and with bilateral CAI (n=11) we tested the ankle with a lower CAIT score.”. It now reads: (See page 4, line180-182).
Line180-182:
“For the non-CAI group, the testing foot was randomly selected, and for CAI group, with unilateral CAI (n=10) we tested the affected ankle, and with bilateral CAI (n=11) we tested the ankle with a lower CAIT score.”
Point 5: Did subjects always know which foot was going to land on the angled platform? If I understand the protocol correctly, the angle was randomized, altered from trial to trial, but never from side to side. Seems like this could introduce some anticipation bias if they always knew which side had the angled surface.
Response 5: Thank you. All participants knew which foot would land on the angled platform, and the ankle tested was consistent throughout the experiment. Because the participants did not know what the angle of the angled platform would be on any trial, there was no anticipation bias.
Point 6: How were the subjects blinded to the angle, since it appears they were allowed to have their eyes open. I don’t think asking them to look forward would be enough. I’m also guessing that they knew which height they were dropping from since it looks like this was altered by adding a spacer to the platform they stood on.
Response 6: Participants were required to keep looking forward and this was supervised by the examiner throughout. We had previously checked that it was not possible to see the platform if gaze was forward.
Point 7: Were subjects instructed to land with equal weight on each foot? How was this controlled?
Response 7: Thank you. We have added this point in the Limitation section. (See page 7, line 340-389)
Line 340-389:
“We did not use any instruments to control whether participants had a balanced distribution of weight between the two feet, a feature which should be improved for future study.”
Point 8: Was the examiner blinded to height or load? Might they be biased by knowing the conditions?
Response 8: The examiner was not blinded to height or load, but was blinded to participants’ ankle stability status.
Point 9: Line 149, were data determined to be normally distributed prior to performing an ANOVA?
Response 9: Thank you. We have added the information about normally distributed data in the Data Analysis section, and moved “was conducted” after “ANOVA”. (See page 5, line202-204).
line202-204:
“Given that the data for the two groups were normally distributed, to determine the effect of CAI, landing heights and loads on ankle proprioception, repeated measures ANOVA was conducted on the AUC scores.”
Results
Point 1: Table 1, kg is a unit of mass, not weight.
Response 1: Thank you. We have changed “Weight” to “Mass”. (See page 5, Table 1).
Table 1. Participant demographic information (Mean ± SD)
|
Characteristic |
Group |
Difference between groups |
|
|
CAI |
Non-CAI |
||
|
N |
21 |
22 |
- |
|
Gender |
M10 F11 |
M11 F11 |
- |
|
Age (y) |
23.4 ± 3.2 |
24.1 ± 2.1 |
t =-0.804, p = 0.426 |
|
Height (cm) |
171.3 ± 8.2 |
169.1 ± 6.3 |
t = 0.972, p = 0.337 |
|
Mass (kg) |
65.6 ± 11.6 |
64.6 ± 9.0 |
t = 0.268, p = 0.790 |
|
CAIT score |
15.6 ± 4.9 |
28.6 ± 1.8 |
t =-11.606, p = 0.000 |
SD = standard deviation, CAI = chronic ankle instability, N = Number, M/F = male/female, CAIT = Cumberland Ankle Instability Tool.
Point 2: Are Table 2 and Figure 3 presenting the same data? It should only be presented in one form. The figure is easier to visualize the observed trends and differences.
Response 2: Thank you. As Table 2 and Figure 3 were presenting the same data, we have now removed Table 2.
Discussion
Point 1: Line 190, confusing, what difference observed in previous studies are you referring to?
Response 1: Thank you. To make it clear, we have added the phrase “between CAI and non-CAI” after the “motor output difference”. (See page 7, line294-297).
Line294-297:
“the difference in proprioceptive performance during landing found here may partially explain the motor output difference between CAI and non-CAI observed in previous studies [43,44].”
Point 2: Line 214-231, in your discussion of the lack of a difference between the two loads, since they knew which foot was landing on the angle, it is possible that they adjusted by landing with more of their weight on the horizontal platform. If this was the case, it could explain the lack of differences.
Response 2: Thank you for your suggestion. We have added this point at the end of the fourth paragraph of Discussion section. (See page 7, line331-334).
Line331-334:
“In addition, although we required participants to land evenly, participants might not have achieved a balanced distribution of weight between the two feet, with more weight on the horizontal platform, thus this may have reduced the load on the inverted foot.”
Point 3: Line 239, did you even run statistics looking for differences between sexes?
Response 3: Statistical tests for differences associated with gender were carried out, but the results showed no significant difference across the four conditions.
Point 4: Seems to me that there are a lot more limitations than you list. For example, if they weren’t blinded to height or foot landing on the angle, this would potentially skew your results. It does not appear that you had a means to measure bodyweight distribution between the feet upon landing to determine if they favored the limb that landed on the angle. The person evaluating the landings did not appear to be blinded to height or load condition.
Response 4:
The person evaluating the landings made no verbal comments to the participants during testing, but checked that gaze was held directly forward.
We have added the point of bodyweight distribution in the Limitations section. (See page 7, line 321-323)
Line 321-323:
“We did not use any instruments to control whether participants had a balanced distribution of weight between the two feet, a feature which should be improved for future study”
Point 5: Line 240, Was the fact that you had a relatively young population by design? Similarly, did you recruit to ensure a similar number of men and women? How were the subjects recruited?
Response 5: We limited the age of our participants to 18-30 years when recruiting subjects to ensure that their proprioception would not be affected by the factor of age. As this study was conducted primarily within the university and we tried to keep the number of men and women equal when recruiting subjects, the group tended towards young adults and half men/women.
The subjects were recruited through posters placed in the school.
Conclusions
Point 1: You found significant effects of group and drop height. Can you comment on whether the magnitude and/or values obtained are clinically significant and if the differences alter their true risk for an ankle injury?
Response 1: Thank you. The study by Waddington and Adams showed that even a 0.04° increase in inversion uncertainty has the potential to raise the probability of instability from 1.2% to 1.22%. Although this 0.02% increase in instability seems insignificant, it could become a significant influence in the occurrence of injury due to the fact that landings are numerous in sports activities. Therefore, the difference in proprioception between the CAI and non-CAI in this study has significant implications for ankle stability and may raise the risk of sprain. We have added this point in the discussion to address this issue. (See page 7, line288-294).
Line288-294:
“According to Waddington and Adams' research [45], even a 0.04° increase in inversion uncertainty has the potential to raise the probability of injury when landing on the inverted ankle from 1.2% to 1.22%. Although this 0.02% increase in injury seems low, it could become a significant influence in the occurrence of injury due to the fact that landings are numerous in sports activities. Therefore, the difference in proprioception between the CAI and non-CAI participants in this study has significant implications for ankle stability and may raise the risk of sprain.”

Reviewer 3 Report
The topic is one of importance given the high number of presentations to health services that are related to concerns in individuals with chronic ankle instability. Also, this is an interesting aim with the study was to investigate how a challenging, repetitive task affects the muscle activities of the lower extremities. I think it would be a more clear study if the following parts were revised and supplemented. These will be discussed below relative to the information of the manuscript.
General Comments:
Overall the manuscript is generally well written and is a topic of interest. However after reading it a number of times I am still left without key take-home points. I believe these points are in the results they just need to be developed.
Specific comments:
Abstract:
1) The aim of this study was to investigate how a was to test the ankle inversion proprioception of individuals with and without chronic ankle instability when they landing at different heights with different loads. This seems like too much of an over simplification of what was done. I do feel that it would be beneficial to explain what specifically you are looking at in relation to the impact of the chronic ankle instability (this also applies to the main body of the paper). Is it the development of the ankle muscles and the impact of chronic ankle instability literature. This needs to be made clearer throughout the paper. (Major Compulsory Revision)
Introduction
2) The first paragraph should have a sentence or two added that better outlines why this study is important related with the chronic ankle instability and the comparison of distal and proximal lower limb muscle activity patterns during external perturbation in subjects with and without functional ankle instability https://pubmed.ncbi.nlm.nih.gov/28843163/ and the impact ankle dorsiflexion range of motion https://pubmed.ncbi.nlm.nih.gov/28070457/
(Major Compulsory Revision)
The authors do a poor job on reviewing relevant literatura related with importance of foot and muscles functional ankle instability https://pubmed.ncbi.nlm.nih.gov/32709515/
3) In the last paragraph, the significance of the proposed word should be included highlighting why your work is important. what is the scientific contribution of this paper? it is not clear how this paper can make a significant contribution to the state of the art. (Major Compulsory Revision).
4) This methods section is poor, needs to present a better rationale for the study and the methodology employed. Also, neither appear information related with inclusion and exclusion criteria, dates, protocol. The study design is a experimental research of ramdom sampling method, where the study was conducted in the hospital or in the university center? This research adhere to reporting STROBE guidelines? (Major Compulsory Revision).
6) Where the experiments carried out? In a hospital? In an educational institution? Provide this information.
7) Add a study flow chart for the readers. (Major Compulsory Revision).
8) The Discussion section is a rehashing of the results. It does not appear that the authors include much interpretation of what the study findings mean for clinical practice or research. (Major Compulsory Revision)
FInally, the conclusión is weak and too long. (Major Compulsory Revision)
Author Response
Response letter
Dear Editor,
Thank you for sending reviewer comments, and thank you for the opportunity to revise this manuscript. We have addressed each reviewer point and revised the manuscript accordingly. We hope that the manuscript is now satisfactory and ready for publication in the Bioengineering.
Response to Reviewer 2 Comments
Point 1: Abstract: The aim of this study was to investigate how a was to test the ankle inversion proprioception of individuals with and without chronic ankle instability when they landing at different heights with different loads. This seems like too much of an over simplification of what was done. I do feel that it would be beneficial to explain what specifically you are looking at in relation to the impact of the chronic ankle instability (this also applies to the main body of the paper). Is it the development of the ankle muscles and the impact of chronic ankle instability literature. This needs to be made clearer throughout the paper. (Major Compulsory Revision)
Response 1: Thank you for your suggestion. We have added “may be important, but are still unclear.” in the Abstract (See page 1, line17-18), and emphasized the relation with, and meaning of, CAI throughout manuscript.
Line17-18:
“The effect of landing heights and loads on ankle inversion proprioceptive performance in individuals with or without chronic ankle instability (CAI) may be important, but are still unclear.”
Point 2: Introduction: The first paragraph should have a sentence or two added that better outlines why this study is important related with the chronic ankle instability and the comparison of distal and proximal lower limb muscle activity patterns during external perturbation in subjects with and without functional ankle instability https://pubmed.ncbi.nlm.nih.gov/28843163/ and the impact ankle dorsiflexion range of motion https://pubmed.ncbi.nlm.nih.gov/28070457/
Response 2: Thank you for your suggestion and recommended literature. We have added “neuromuscular control is deficient and” before the “ankle inversion proprioception to be significantly impaired” in the first paragraph. Accordingly, we also added “and the lower limb proximal muscle activity pattern is altered” in front of “in individuals with CAI” and the literature recommended has been cited. (See page 1, line39-42)
Line39-42:
“Studies have shown that neuromuscular control is deficient and ankle inversion proprioception to be significantly impaired and the lower limb proximal muscle activity pattern is altered in individuals with CAI. [10,11]”
Reference:11. Kazemi, K.; Arab, A.M.; Abdollahi, I.; López-López, D.; Calvo-Lobo, C. Electromiography comparison of distal and proximal lower limb muscle activity patterns during external perturbation in subjects with and without functional ankle instability. Hum Mov Sci 2017, 55, 211-220, doi:10.1016/j.humov.2017.08.013.
We have added “, which may lead to” before the “recurrent ankle sprains” and new references were inserted here. (See page 2, line86)
Line86:
“CAI is a condition characterized by pain, weakness, reduced ankle range of motion, perceived ankle ‘giving way’ sensation, proprioception deficit, which may lead to recurrent ankle sprains [25-28].”
Reference:27. Romero Morales, C.; Calvo Lobo, C.; Rodríguez Sanz, D.; Sanz Corbalán, I.; Ruiz Ruiz, B.; López López, D. The concurrent validity and reliability of the Leg Motion system for measuring ankle dorsiflexion range of motion in older adults. PeerJ 2017, 5, e2820, doi:10.7717/peerj.2820.
- Cruz-Díaz, D.; Lomas Vega, R.; Osuna-Pérez, M.C.; Hita-Contreras, F.; Martínez-Amat, A. Effects of joint mobilization on chronic ankle instability: a randomized controlled trial. Disabil Rehabil 2015, 37, 601-610, doi:10.3109/09638288.2014.935877.
Point 3: Introduction: The authors do a poor job on reviewing relevant literatura related with importance of foot and muscles functional ankle instability https://pubmed.ncbi.nlm.nih.gov/32709515/
Response 3: Thanks for your suggestion. We have added the following information in manuscript. (See page 2, line87-90)
“Although the soft tissues of the foot (e.g. fascia, muscles) perform a crucial role in maintaining the ankle stability, recurrent ankle sprains disrupt the original structure. Studies have shown that the thickness of the plantar fascia is reduced after lateral ankle sprain [29].”
Reference:29. Romero-Morales, C.; López-López, S.; Bravo-Aguilar, M.; Cerezo-Téllez, E.; Benito-de Pedro, M.; López López, D.; Lobo, C.C. Ultrasonography Comparison of the Plantar Fascia and Tibialis Anterior in People With and Without Lateral Ankle Sprain: A Case-Control Study. J Manipulative Physiol Ther 2020, 43, 799-805, doi:10.1016/j.jmpt.2019.11.004.
Point 4: Introduction: In the last paragraph, the significance of the proposed word should be included highlighting why your work is important. what is the scientific contribution of this paper? it is not clear how this paper can make a significant contribution to the state of the art. (Major Compulsory Revision).
Response 4: Thank you. We have clarified the details about the significance of this study at the end of the first paragraph. (See page 1, line 45-60).
Line 45-60:
“Thus, exploring ankle inversion proprioception during landing is essential to understand the sensorimotor mechanisms underlying ankle sprains, and may inform prevention and rehabilitation of ankle sprain.”
Point 5: This methods section is poor, needs to present a better rationale for the study and the methodology employed. Also, neither appear information related with inclusion and exclusion criteria, dates, protocol. The study design is a experimental research of random sampling method, where the study was conducted in the hospital or in the university center? This research adhere to reporting STROBE guidelines? (Major Compulsory Revision).
Response 5: Thanks for your suggestion. Accordingly, we have revised the rationale of the study and adapted the methodology section according to the reporting STROBE guideline. (See page 2-5, line 93-176)
We have now added the “Study design” section in front of the “Participants”, which noted the dates. (See page 2, line 107-110)
Line 107-110:
“2.1. Study design
A cross-sectional study was performed between June and July 2021 according to the Strengthening the Reporting of Observational Studies in Epidemiology (STROBE) recommendations.”
We have included the details about the inclusion and exclusion criteria (See page 3, line 128-138).
Line 128-138:
“the CAI participants must have (i) had at least one ankle sprain that caused inflammatory reaction (pain, swelling, etc) in the previous 12 months; (ii) at least two episodes of ankle instability "giving way" or repeated sprain within 6 months before the test; (iii) a Cumberland ankle instability tool score (CAIT) [39]<24 and (iv) not had an ankle injury within 3-months of being tested. These inclusion criteria align with the recommendations of the International Ankle Consortium [39]. The non-CAI participants with no subjective reports of ankle instability, recurrent ankle sprains, neurological or motor dysfunction. All participants were excluded from the study if they had history of lower limb surgery, fracture; or any acute injury of other joints of the lower limb in the 3-months prior to study commencement.”
To make it clearer, we have now added “inclusion” in front of “criteria” in the Methods – Participants (See page 3, line 133). It now reads:
“These inclusion criteria align with the recommendations of the International Ankle Consortium.”
We have mentioned that the study was conducted in a university laboratory (See page 4, line 179).
Line 179:
“The experiment was conducted in a university laboratory.”
Point 6: Where the experiments carried out? In a hospital? In an educational institution? Provide this information.
Response 6: Thank you. We have mentioned that the experiments was conducted in a university laboratory (See page 4, line 179).
Line 179:
“The experiment was conducted in a university laboratory.”
Point 7: Add a study flow chart for the readers. (Major Compulsory Revision).
Response 7: Thank you for your suggestion. We have added the following flow chart in manuscript. (See page 5, line 192-194)
Figure 3. The flow chart of this study. 100% meant original body weight and 110% meant 10% extra body weight; 10cm and 20cm indicated landing heights)
Point 8: The Discussion section is a rehashing of the results. It does not appear that the authors include much interpretation of what the study findings mean for clinical practice or research. (Major Compulsory Revision)
Response 8: Thank you for your suggestion. We have revised the discussion and have placed much more emphasis on our own result and its clinical meaning. We have now revised the discussion.
Point 9: Finally, the conclusion is weak and too long. (Major Compulsory Revision)
Response 9: Thank you. We have now revised the discussion.
“4. Discussion
Consistent with our hypothesis, we found that the overall ankle inversion proprioceptive performance of CAI patients was significantly worse than that of non-CAI participants, suggesting that CAI patients have impaired proprioceptive control when landing on uneven surface. In addition, ankle proprioceptive discrimination sensitivity was significantly worsened by increased landing height, but not loading (Figure 4), which was true for both CAI and non-CAI groups.
Our results showed that the proprioceptive acuity of CAI participants was significantly worse than those without CAI across the 4 different testing conditions. These findings are consistent with prior studies [12,41] and further supports the notion that CAI patients have impaired somatosensory control during landing on inverted ankle [12]. Some studies have shown that individuals with CAI exhibit altered peak proximal muscle forces, force-generating capacities, as well as greater hip flexion and ankle inversion angles and peak vertical ground reaction forces during landing tasks [42-44]. Our findings complement the substantial deviations in the lower limb motor output observed between CAI and non-CAI individuals and have shown that ankle proprioceptive input is also different between the two groups. According to Waddington and Adams' research [45], even a 0.04° increase in inversion uncertainty has the potential to raise the probability of injury when landing on the inverted ankle from 1.2% to 1.22%. Although this 0.02% increase in injury seems low, it could become a significant influence in the occurrence of injury due to the fact that landings are numerous in sports activities. Therefore, the difference in proprioception between the CAI and non-CAI participants in this study has significant implications for ankle stability and may raise the risk of sprain. Given that ankle proprioception is fundamental for lower limb motor control [23], the difference in proprioceptive performance during landing found here may partially explain the motor output difference between CAI and non-CAI observed in previous studies [43,44]. Future research may explore if any rehabilitation program targets ankle proprioception [46] may have positive effects on lower limb motor control in people post ankle sprain that may or may not develop CAI.
In terms of the effect of landing height on lower limb biomechanics, research has shown that with increasing landing height, the ankle dorsiflexion, knee extension and peak ankle plantarflexion moments are significantly altered in individuals with CAI [47]. The results of the current study show that increased landing height could impair ankle inversion proprioception in both CAI and non-CAI participants. This evidence supports the notion that the larger movement amplitude, the worse proprioceptive performance [48], suggesting that landing from a higher place may be associated with increased noise in the perceptual systems of brain [48,49] so that ankle inversion movement discrimination sensitivity is decreased. Previous upper [48,50] and lower [51] limb proprioception studies have found that larger movement resulted in worse proprioceptive acuity. These are consistent with the findings of our result. A greater movement amplitude would generate more noise and increased uncertainty when judging limb positions in space. This finding suggests that higher jump-landing may increase the risk of ankle injury and partially explains why landing from a jump is one of the most common mechanisms for ankle injury [2], especially in the ankle joint inversion landing conditions [52].
However, we found extra 10% of body weight for the proprioceptive measurement did not differ from 100% body weight conditions. This evidence is contrary to the argument that ankle proprioception simply relied on mechanoreceptors around the foot-ankle complex [53]. Han et al. [9,54] argue that proprioception assessment methods can be classified into testing passively “imposed” and actively “obtained” proprioception. Imposed methods [53] believed that proprioception is completely reliant on information received from the peripheral proprioceptors mechanoreceptors (i.e. muscles, joints, and skin) [55]. In contrast, actively obtained methods [56] were developed on the bases of ecological validity concept that proprioceptive performance is not fully determined by passively imposed proprioceptive signals from mechanoreceptors, but requires adaptive central processing of multiple sources of information [9]. If the “imposed” proprioception view is true, then both increased landing height and load could further activate mechanoreceptors located around the foot-ankle complex, improving ankle proprioception. However, the results here did not support this notion. The possible reason is that additional 10% of load may not have been large enough, and the central nervous system may have mechanisms to adapt to the noise generated by mild changes in weight-bearing condition. In addition, although we required participants to land evenly, participants might not have achieved a balanced distribution of weight between the two feet, with more weight on the horizontal platform, thus this may have reduced the load on the inverted foot.
This study compared ankle inversion proprioception during landing in both CAI and non-CAI populations in the face of height and loading conditions. One of the limitations of this study was that we did not collect data about the physical activity and occupation of the participants included in this study, which may have an impact on the results. Furthermore, we did not quantify kinematic patterns so that precise changes in movement extent of the participants at different heights and loading states to be observed. We did not use any instruments to control whether participants had a balanced distribution of weight between the two feet, a feature which should be improved for future study. In addition, previous research has shown that female and male individuals with CAI performed differently on a range of functional tasks [57]. However, given relatively small sample size of the current study, the sex differences in this proprioceptive task were not revealed. Furthermore, the participants involved in the current study were relatively young and it is unknown the findings here can be generalized to other age groups.”

Round 2
Reviewer 1 Report
Authors addressed all my comments
Reviewer 3 Report
In their first revision of manuscript, the authors have addressed my questions/comments properly.